# Linear Dimensional Change and Ultimate Tensile Strength of Polyamide Materials for Denture Bases

**DOI:** 10.3390/polym13193446

**Published:** 2021-10-08

**Authors:** Bozhana Chuchulska, Stefan Zlatev

**Affiliations:** 1Department of Prosthetic Dental Medicine, Faculty of Dental Medicine, Medical University of Plovdiv, 4000 Plovdiv, Bulgaria; bogana_68@abv.bg; 2CAD/CAM Center of Dental Medicine, Research Institute at the Medical University of Plovdiv, 4000 Plovdiv, Bulgaria

**Keywords:** denture base materials, polyamide, injection molding, dimensional stability, ultimate tensile strength, thermocycling

## Abstract

The aim of the current study was to evaluate the dimensional changes and ultimate tensile strength in three polyamide materials for denture bases fabrication through injection molding, subjected to artificial aging and different storage conditions. A total of 333 test specimens fabricated from Biosens (BS; Perflex, Netanya, Israel), Bre.flex 2nd edition (BF; Bredent, Senden, Germany) and ThermoSens (TS; Vertex Dental B.V., Soesterberg, The Netherlands)—*n* = 111 per material—were equally divided into three groups (*n* = 37) based on different treatments and storage conditions. Test samples allocated to the “Control group” were not artificially aged and stored in water for 24 h. Both “Treatment 1 group” and “Treatment 2 group” were subjected to thermocycling, the former dehydrated and the latter stored in water between cycle-sets. Linear changes and ultimate tensile strength were measured and analyzed for storage condition and material influence on the outcome variables. A Welch ANOVA test with Games–Howell post-hoc analysis was used to compare the influence of treatments across different materials. Significant differences were found for all three included materials with *p* values ranging from <0.05 to <0.001 for linear dimensional changes. The magnitude of alterations varied and was large for BS (Perflex, Israel) (ω^2^ = 0.62) and BF (Bredent, Germany) (ω^2^ = 0.47) and small but significant for TS (Vertex Dental B.V., The Netherlands) (ω^2^ = 0.05). However, results seem to fall into clinically acceptable range. Significant differences were also observed for the ultimate tensile strength test with the same range of *p*-values. All three materials showed different initial ultimate tensile strengths and varying reaction to artificial aging and storage with the lowest alterations observed for BF (Bredent, Germany) (ω^2^ = 0.05). Within the limitation of this study, it can be concluded that all three materials show different dimensional and mechanical properties when subjected to artificial aging and different storage. Although linear dimensions show significant changes, they seem to be clinically irrelevant, whereas the change in ultimate tensile strength after only 6-month equivalent clinical use was substantial for BS (Perflex, Israel) and TS (Vertex Dental B.V., The Netherlands).

## 1. Introduction

Polymer materials were introduced in dentistry earlier than in any other healthcare specialty. The first acrylic resin for dental purposes was registered under the name Palapont in 1943 by Kulzer in Germany. By the middle of the 20th century, they became the material of choice for fabrication of denture bases, maxillofacial prostheses, orthodontic and other dentistry-related appliances [1]. Acrylic-based resins have low density and thermal conductivity, good resistance to chemical solvents as well as color and appearance, which closely mimic oral tissues. The most frequently used acrylic resins in dentistry to date are with thermal activation of the polymerization process [1,2]. Despite their many advantages a number of shortcomings, namely dimensional stability issues, lack of sufficient mechanical and wear resistance properties, potential allergenic and cytotoxic effects among others, have led to the development and introduction of other materials for denture base fabrication [1,3].

In 1950, polyamide polymers were introduced as an alternative to acrylic-resin materials. They are thermoplastic polymers that can become highly elastic through controlled heating. In 1956, the society of artificial organs created a special group of medical-grade thermoplastic polymers, which triggered substantial research towards their potential use in medicine. In 1953, Arpad and T. Nagi documented the first use of thermoplastic resins for dentures manufacturing in their laboratory in New York. Several years later, in 1959, both researchers founded the company Valplast. These events marked the start of thermoplastic resin usage in dentistry with more than 90 countries currently using these materials for denture fabrication [1].

The oral cavity is considered a complex environment that subjects dental materials to an array of challenges. Its relative humidity above 90%, constant contact of restorations to saliva, cold/hot air, food and liquids (as well as enzymes, bacteria and varying pH levels) can severely affect restoration’s color stability, physical and mechanical properties [4]. Dental materials used for denture base fabrication are prone to water sorption. The latter is especially true for acrylic resins, where the sorption amount is proportional to the resin components with high polarity. Acrylic resins tend to form hydrogen bonds with water molecules, breaking weaker interchain bonds and deteriorating physical and mechanical properties of the prostheses [5,6]. First generation of polyamide materials had inferior properties regarding deformation, water sorption and resistance to fatigue from cyclic thermal changes and mechanical loading [4]. This group of materials is characterized by hydrophilic amide bonds that form the resin’s main chains which makes them prone to water sorption. Manufacturers claim that in their later generations of products water sorption and its associated mechanical and dimensional shortcomings are eliminated through amide group concentration control.

Dimensional stability and mechanical properties of polyamide materials for denture bases have been previously tested. Most of the studies focus on comparing different classes or brands of materials for denture base fabrication [7,8,9,10]. Testing conditions varied considerably based on the research objective and ranged from conducting the experiments immediately after test specimens’ fabrication, to a time point following water immersion or dehydration [11]. Several studies considered artificial aging in combination with different storage conditions including cyclic dehydration. Artificial aging, hydration and dehydration cycles seem to significantly deteriorate mechanical properties of denture base materials, but this influence is considered more pronounced for acrylic-based resins and less so for polyamides [12]. Only one article was identified to investigate ultimate tensile strength, but this was done without artificial aging of the test specimens [8].

The aim of the current study was to evaluate the dimensional changes and ultimate tensile strength in three polyamide materials for denture bases fabricated trough injection molding (Biosens—BS (Perflex, Israel), Bre.flex 2nd edition—BF (Bredent, Germany) and ThermoSens—TS (Vertex Dental B.V., The Netherlands)) subjected to artificial aging and different storage conditions (no artificial aging—control, artificial aging with dehydration—test group 1; artificial aging without dehydration—test group 2). In order to assess the influence of treatment and storage conditions on the outcome variables the following hypothesis was tested:
H_0_—different storage conditions and artificial aging will not significantly influence the dimensions and tensile strength of the test samples.H_a_—different storage conditions and artificial aging will significantly influence the dimensions and tensile strength of the test samples.

## 2. Materials and Method

### 2.1. Sample Size Calculation and Study Design

An a-priori power analysis with pre-set *p*-value of 0.05 and power of 0.80 with a 3-group balanced design was performed using G*Power. The analysis determined a total of 111 required specimens for each material (*n* = 37 per group) in order to satisfy the aforementioned parameters.

The design of the study, including the tested groups and their respective treatments, is presented on Figure 1.

### 2.2. Specimen Fabrication

Three polyamide materials were tested for dimensional stability and tensile strength. The corresponding fabrication parameters, as well as the machines used in the study are presented in Table 1.

A flask from aluminum alloy (EN AW—7075—AlZnMgCu1.5; W-Nr 3.4365) for simultaneous fabrication of four test specimens was designed and milled for the study. The sprue channel was designed with a 10 mm diameter and additional air-flow channels were added to facilitate the manufacturing process. The test specimens were designed with a “hour-glass” shape and predefined dimensions presented in Figure 2.

The specimens produced from each material were fabricated following their respective manufacturers’ instructions. A total of 84 injection cycles were performed yielding the required 333 specimens.

Test specimens from each material were divided into three equal groups (*n* = 37):
Control—without treatment (C)Test group 1—with artificial aging and dehydration (T1)Test group 2—with artificial aging and no dehydration (T2).

### 2.3. Treatment and Storage

Test samples from the control group did not receive any additional treatment after fabrication. They were separated from the flask and stored in an airtight glass container without access of direct sunlight at room temperature for 24 h before diameter measurements and tensile strength testing. Thermocycling was used to artificially age the samples from T1 and T2 using LTC 100(LAM Technologies, Firenze, Italy). Consequential immersion in two tubs with temperatures 5 °C and 55 °C for 30 s each, and a 30 s “drying” period between tubs, were performed for all samples in both groups. In total, 20 thermocycling sets, 250 cycles each, were conducted making a total of 5000 cycles. Airtight glass containers, under no direct lighting, at room temperature were used for all samples. In T1, a dry container was used, whereas the specimens from T2 were immersed in distilled water.

### 2.4. Measurement of Linear Differences and Tensile Strength

Two calibrated investigators—B.CH. and S.Z.—independently performed the measurements of the testing area (midpoint of the hourglass) for each sample. A digital caliper Mannesmann (Düsseldorf, Germany) was used. Before measuring each specimen, the instrument was calibrated according to the manufacturer’s instructions. All test samples were measured three times at the midpoint of their testing area, values were averaged and recorded. A difference larger than 0.05 mm between the two investigators rendered the measurement erroneous and the procedure was repeated for the given sample. Ultimate tensile strength was tested in LMT (Lam Technologies, Italy) with a speed of 0.5 mm/s until specimen failure. The measured values were recorded in newtons, subsequently transformed in MPa, and included for analysis.

### 2.5. Statistical Analysis

The data was statistically processed using R [13]. Descriptive statistics were used to characterize the variables of interest. A sample *t*-test was used to assess whether the manufacturing procedure affects the test sample’s diameters. Welch’s ANOVA with Games–Howell post-hoc analyses were used to test the hypothesis for significant differences of treatment and material’s influences on the outcome variables.

## 3. Results

Results are presented in two subsections. The division criteria used was based on the received treatment and storage conditions and outcome variable of interest—dimensional changes and ultimate tensile strength. Since the assumption of homogeneity of variance was not met for our results, we used the Welch’s adjusted F test with the Games–Howell post-hoc analysis. Summary statistics for the diameter and ultimate tensile strength measurements—means and standard deviations for each material across different treatment and storage conditions, are presented in Figure 3 and Figure 4, respectively.

### 3.1. Dimensional Accuracy of Test Samples across Different Treatments

The Welch’s F test revealed a significant main effect for all included materials (BS—F(2, 68.7) = 94.2, *p* < 0.001; BF—F(2, 70.4) = 50.4, *p* < 0.001, TS—F(2, 66.5) = 4.17, *p* < 0.05), indicating that not all test samples subjected to different treatment and storage conditions had the same average diameter. A sample *t*-test was conducted to determine whether there were significant deviations from the predefined test samples diameter (1.5 mm) for samples produced with the three included materials, without them being subjected to additional treatment—C-group. The results showed significant differences for all three materials within this treatment and storage condition: BS—t(36) = 4.29, *p* < 0.001; BF—t(36) = 11.1, *p* < 0.001; TS—t(36) = 23, *p* < 0.001.

Furthermore, the constructed interaction plot presented in Figure 4 revealed different reaction patterns across the materials, indicating possible interactions between materials and treatment conditions. Specimens fabricated from BS exhibited an increase in diameter when subjected to T1 and T2 conditions, whereas BF showed decrease from the values obtained for the control group. Dehydration did not seem to affect TS. Nevertheless, a marked increase in diameter is observed in the T2 group.

The estimated general effect size for BS—omega squared (ω^2^ = 0.62) indicated that approximately 62% of the total variation in the measured test samples’ diameters is attributable to differences between treatment and storage conditions. The conducted Games-Howell post-hoc analysis revealed that all tested pairs differed significantly at the 0.001 level with large effect sizes—Table 2.

The estimated general effect size for BF—omega squared (ω^2^ = 0.47) indicated that approximately 47% of the total variation in the measured diameters is attributable to differences between treatment and storage conditions. The post-hoc analysis results are presented in Table 3. Significant differences were observed between the pairs C-T1 and T1-T2 at the 0.001 level with large effect sizes.

The estimated general effect size for TS—omega squared (ω^2^ = 0.05) indicated that approximately only 5% of the total variation in the measured diameters is attributable to differences between treatment and storage conditions. The post-hoc analysis results are presented in Table 3. Significant differences were observed between the pairs C-T2 and T1-T2 at the 0.05 level with moderate effect sizes. The artificial aging and dehydration (T1) do not seem to lead to significant diameter change when compared to the control group.

### 3.2. Ultimate Tensile Strength of Test Samples Manufactured from the Same Material across Different Treatments

The Welch’s F test revealed a significant main effect for all included materials (BS—F(2, 61.91) = 182, *p* < 0.001; BF—F(2, 68.3) = 4.23, *p* < 0.05, TS—F(2, 61.5) = 144, *p* < 0.001), indicating that not all test samples subjected to different treatment and storage conditions showed the same ultimate tensile strength.

The analysis of the interaction plot presented in Figure 4 indicates a sharp decrease in the ultimate tensile strength results for BS in both artificial aging and storage conditions (T1 and T2) and a less pronounced but significant decrease for test specimens manufactured from TS. Dehydration does not seem to influence the outcome variable for samples in the BF group. The T2 condition led to a decrease in the ultimate tensile strength for these specimens, but with a moderate effect size.

The estimated general effect size for BS—omega squared (ω^2^ = 0.76) indicated that approximately 76% of the total variation in the measured test samples’ ultimate tensile strength is attributable to differences between treatment and storage conditions. The conducted Games–Howell post-hoc analysis revealed that all tested pairs differed significantly at the 0.001 level with large effect sizes—Table 3.

The estimated general effect size for BF—omega squared (ω^2^ = 0.05) indicated that approximately 5% of the total variation in the measured ultimate tensile strength is attributable to differences between treatment and storage conditions. The post-hoc analysis results are presented in Table 3. Significant differences were observed only in the pair C-T2 at the 0.05 level with a moderate effect size.

The estimated general effect size for TS—omega squared (ω^2^ = 0.72) indicated that approximately 72% of the total variation in the measured diameters is attributable to differences between treatment and storage conditions. The post-hoc analysis results are presented in Table 3. Significant differences were observed between all tested pairs at the 0.001 level with large effect sizes.

## 4. Discussion

A non-anatomic, hour-glass shaped test specimen was chosen in order to eliminate anatomical and thickness variability of a denture-like sample and to facilitate the subsequent ultimate tensile strength test. The dimensions of the narrow part of test specimens were designed to closely resemble real clinical conditions based on recommendations for average reported values in non-metal clasps and denture base thickness [14,15]. Furthermore, the constructed matrix from aluminum alloy eliminates inconsistency that may occur due to the type of dental stone, wax pattern, and their associated technical processing during specimen fabrication. Using a simple shape instead of anatomically based one simplifies the linear measurements. Thus, the specimen’s dimensional changes could be directly attributed to the material and the processing method [16]. The technique of using digital caliper has the advantage of being simple and easily available but has some limitations. It has been reported that the force applied by the operator might influence the resulting measurements contributing to minor errors [17]. In the current study, an attempt was made to compensate for the aforementioned by making 6 total measurements for each test specimen, split evenly (*n* = 3) across two independent operators, and taking the average result.

Cyclic thermal and masticatory load, as well as storage conditions, have a substantial effect over different mechanical and physical properties of materials used to fabricate partial and full denture bases. Artificial aging simulates real clinical conditions with 10,000 cycles of thermocycling approximating a period of one year [18,19]. In the current study, test specimens from T1 and T2 group were subjected to a total of 5000 cycles that represent six-months clinical usage. Series of 250 cycles each were carried out with a rest period of 24 h during which samples from each corresponding group were stored in a dry environment or immersed in water. This approach closely resembles real clinical conditions where a denture is used, then removed and stored in a dry or wet container during the night.

Results obtained for linear dimensional changes for the three materials under condition “C” ranged between ±3%. BS showed the smallest change within this group (−1.3%), whereas the other two materials exhibited equal diameter variations (3%) but with opposite directions—specimens fabricated from TS shrunk and the samples from BF expanded. Several publications in the specialized literature investigate linear dimensional changes but are limited to acrylic based materials or employ test specimen dimensions and/or measuring methodology that is different than those used in the current study [20,21,22,23]. These reasons restrict the possibility of direct comparison between our findings and the results obtained by others. Nevertheless, the range of linear changes resulting from material processing and 24-h storage in water seems to be within clinically acceptable limits, also reported in part of the aforementioned articles for acrylic based materials [16]. Additionally, a linear change might not fully represent the complex pattern of dimensional alterations that occur during fabrication of dentures, hence its usefulness in translating the results to real clinical situations is somewhat questionable [22,23].

Artificial aging through thermocycling and storage conditions significantly influenced the linear dimensions for TS, BF and BS obtained in our study. Although the results are considered within a clinically acceptable norm, the pattern of change and different behavior of the included materials is noteworthy [16]. Samples produced from TS showed the least linear deformation following expected pattern—reduction after dehydration, albeit insignificant compared to the “C” group, and expansion under the “T2” condition [9]. An interesting reaction to artificial aging and the two modalities of storage are observed in the BS group. Regardless of “T1” and “T2”, samples produced from this material expanded significantly. BF test specimens had a similar pattern to TS but with larger values for percent-linear change. Despite that, differences between groups “C” and “T2” were insignificant. These results suggest that BF is dimensionally sensitive to storage conditions.

Ultimate tensile strength is related to the flexural properties of the injection-molded polyamide materials for denture bases. They are of significant clinical importance for non-metal clasp dentures. The change in flexibility of the clasps might affect the ease of insertion and removal of the RPD and the stress to the abutment teeth [2]. Ultimate tensile strength has been previously investigated for acrylic- and polyamide-based materials for denture base fabrication [8]. However, the combined effect of water sorption and thermal cycling on ultimate tensile strength has not been studied for the three materials analyzed in this study. A significant difference was observed between BS, BF and TS in all the treatment and storage groups. Furthermore, results indicate significant deterioration in the mechanical properties for the specimens, most noticeable in the BS group. Although BF specimens were the most dimensionally unstable of the three, test samples produced from this material showed the least decline in ultimate tensile strength. These findings suggest that the resistance to fracture and flexibility of clasps in polyamide-based dentures will change with their usage over time. A possible outcome might be a necessity for more control appointments, even reduced period of usage, and decreased retention if a permanent deformation in the retentive elements occurs.

## 5. Conclusions

Within the limitation of the current study the following conclusions can be drawn:The diameter of test samples within the control group fabricated from all three materials differed from the predefined values in the matrix form and from each other, which suggests that the manufacturing process and type of material has a significant influence on their dimensions.When subjected to the predefined treatment and storage conditions, the three included materials react in completely different ways, showing significant changes. However, TS seems more dimensionally stable compared to the other two, after artificial aging and different storage conditions.Artificial aging and storage conditions contribute to significant decrease in the ultimate tensile strength for all tested materials. BF experienced the least decrease in ultimate tensile strength after artificial aging, regardless of the storage method.

## Figures and Tables

**Figure 1 polymers-13-03446-f001:**
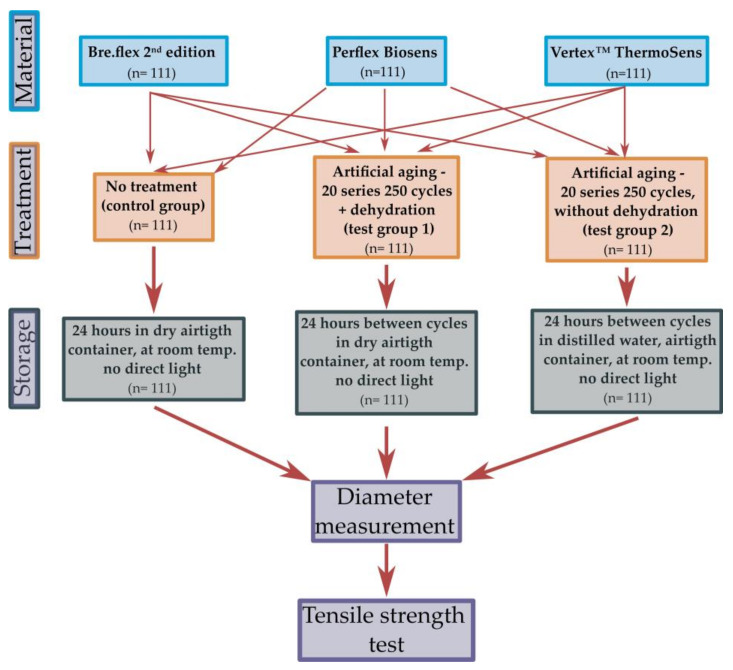
Study design flow-chart.

**Figure 2 polymers-13-03446-f002:**
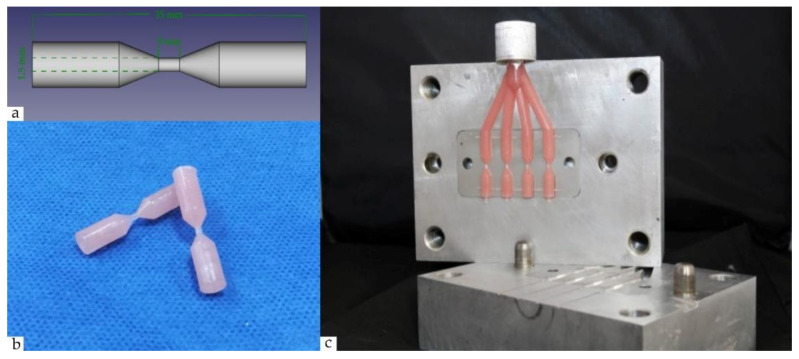
Test specimens’ fabrication: (**a**) design; (**b**) finished test samples, (**c**) matrix form with injected specimens.

**Figure 3 polymers-13-03446-f003:**
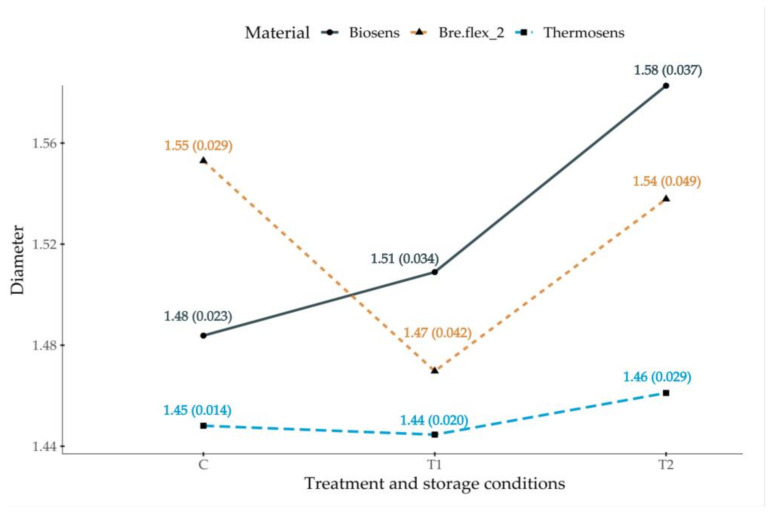
Treatment and storage conditions, and material as diameter predictors. Mean and standard deviation values in brackets () are presented for each group.

**Figure 4 polymers-13-03446-f004:**
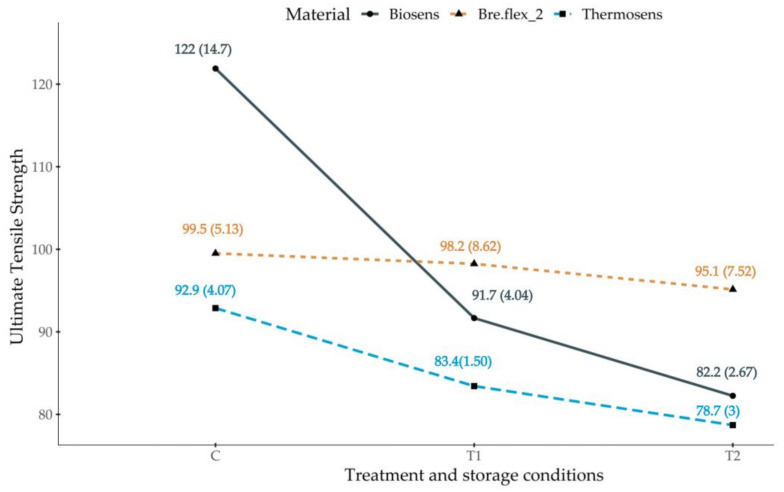
Treatment and storage conditions, and material as ultimate tensile strength predictors. Mean and standard deviation values in brackets () are presented for each group.

**Table 1 polymers-13-03446-t001:** Materials, type, manufacturer, and fabrication parameters included in the study.

Material	Type	Time	Temperature	Pressure	System	Manufacturer
Bre.flex 2nd edition (BF)	Polyamide (PA-12)	15 min	280 °C	7.5 Bar	Thermopress 400	Bredent, Germany
Perflex Biosens (BS)	Polyamide (MSDS: no declaration)	18 min	300 °C	8–9.5 Bar	Thermopress 400	Perflex, Israel
Vertex^TM^ ThermoSens (TS)	Polyamide (MSDS: no declaration)	18 min	290 °C	6 Bar	Vertex Thermoject 22	Vertex Dental B.V., The Netherlands

**Table 2 polymers-13-03446-t002:** Games–Howell post-hoc test for diameter by treatment and storage conditions for the three tested materials.

Material	Difference of Levels	Difference of Means	Se	95% Confidence Interval		*t*-Value	*p*
				Lower	Upper		
Biosens	C-T1	2.51 × 10^−2^	4.73 × 10^−3^	9.08 × 10^−3^	4.12 × 10^−2^	3.76	***
	C-T2	9.89 × 10^−2^	5.07 × 10^−3^	8.17 × 10^−2^	1.16 × 10^−1^	13.8	***
	T1-T2	7.38 × 10^−2^	5.82 × 10^−3^	5.41 × 10^−2^	9.35 × 10^−2^	8.97	***
Bre.Flex 2nd edition	C-T1	−8.32 × 10^−2^	5.95 × 10^−3^	−1.03 × 10^−1^	−6.31 × 10^−2^	9.89	***
	C-T2	−1.5 × 10^−2^	4.75 × 10^−3^	−3.12 × 10^−2^	9.27 × 10^−4^	2.26	ns
	T1-T2	6.8 × 10^−2^	5.91 × 10^−3^	4.80 × 10^−2^	8.82 × 10^−2^	8.15	***
ThermoSens	C-T1	−3 × 10^−3^	2.83 × 10^−3^	−1.31 × 10^−2^	6.08 × 10^−3^	0.879	ns
	C-T2	1.3 × 10^−2^	3.71 × 10^−3^	3.01 × 10^−4^	2.56 × 10^−2^	2.47	*
	T1-T2	1.65 × 10^−2^	4.09 × 10^−3^	2.62 × 10^−3^	3.04 × 10^−2^	2.85	*

ns—nonsignificant, *—*p* < 0.05, ***—*p* < 0.001.

**Table 3 polymers-13-03446-t003:** Games–Howell post-hoc test for ultimate tensile strength by treatment and storage conditions for the three tested materials.

Material	Difference of Levels	Difference of Means	Se	95% Confidence Interval		*t*-Value	*p*
				Lower	Upper		
Biosens	C-T1	−30.2	1.77	−36.3	−24.2	12.1	***
	C-T2	−39.7	1.73	−45.6	−33.7	16.2	***
	T1-T2	−9.42	0.564	−11.3	−7.51	11.8	***
Bre.Flex 2nd edition	C-T1	−1.26	1.17	−5.23	2.70	0.765	ns
	C-T2	−4.37	1.06	−7.96	−0.778	2.92	*
	T1-T2	−3.11	1.33	−7.61	1.40	1.65	ns
ThermoSens	C-T1	−9.45	0.505	−11.2	−7.72	13.2	***
	C-T2	−14.2	0.588	−16.2	−12.2	17.0	***
	T1-T2	−4.73	0.390	−6.06	−3.40	8.57	***

ns—nonsignificant, *—*p* < 0.05, ***—*p* < 0.001.

## Data Availability

The data presented in this study is available upon request from the corresponding author.

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
