# Peer review of "Linear Dimensional Change and Ultimate Tensile Strength of Polyamide Materials for Denture Bases"

_polymers, 2021, doi:10.3390/polym13193446_

Round 1

Reviewer 1 Report

The article "Linear Dimensional Change and Ultimate Tensile Strength of
Denture Base Polyamide Materials for Injection Molding" is interesting, concise and well organized.

It has to be further checked for correct English and editing errors.

My comments are as follows:

Please give manufacturer and location in brackets each time you mention a material or machine (abstract included).

Please provide information on what basis were the test parameters chosen. 

Most of the references are old. I understand that some of them are basic for the subject, but still, newer ones are needed.

The references should follow the template.  

Author Response

We would like to express our gratitude for the reviewer’s valuable input and comments regarding our manuscript.

Following are the responses to the comments/suggestions:

Comment #1

It has to be further checked for correct English and editing errors.

Response:

A rigorous check for grammatical and editing errors has been performed by a native speaker.

Comment #2

Please give manufacturer and location in brackets each time you mention a material or machine (abstract included).

Response:

Thank you for pointing this out. The abstract has been updated with the location and manufacturers of the included materials.

Comment #3

Please provide information on what basis were the test parameters chosen. 

Response:

Thank you for the valuable comment. In the first paragraph of the discussion section (lines 252-254) an explanation is provided regarding the chosen shape and method for diameter measurement. The following has been added, explaining the dimensions of the testing part of the specimens (line 244-246):

“The dimensions of the narrow part of test specimens were chosen to closely resemble real clinical conditions based on average reported values for non-metal clasps and denture base thickness.”

Another reason for choosing the dimensions of the entire test specimen in the current study is an unpublished simulation study. The aforementioned was conducted when creating the flask for simultaneous production of test specimens to determine a shape with optimal flow/heat conductivity characteristics for the tested materials:

The traction speed of the machine was chosen based on the following recommendation found in section 8.2 in the ASTM D638 – 14 standard for tensile properties of plastics:

“Use the lowest speed for the specimen geometry being used, which gives rupture within 0.5 to 5-min testing time.”

Several speeds were tested with 0.05mm/s (30mm/min) being the lowest producing rupture within the time limit stated in the standard.

Comment #4

Most of the references are old. I understand that some of them are basic for the subject, but still, newer ones are needed.

Response:

Thank you for your comment. To our best abilities we have updated the reference list with more recent publications.

Comment #5

The references should follow the template.  

Response:

Thank you for your valuable comment. All references are formatted according to the requirements.

We would like to thank the dear Reviewer again for the time and effort reviewing the quality of our work. It is our hope that addressing the issues would make the manuscript more readable and increase its usefulness to the research community.

Reviewer 2 Report

This study aimed to evaluate the dimensional changes and ultimate tensile strength in three polyamide denture base materials for injection molding, subjected to artificial aging and different storage conditions. This study is well written, with an excellent introduction to the subject. However, some adjustments are needed.

It would be interesting if the objective of the study presented all the information about the study variables. “three polyamide denture base materials” and “different storage conditions”, insert all variables in parentheses, for example.

The sample number (n) represented in fig 1 is 111 for all groups. I believe that this number represents the universe of each material (N), and the sample for each treatment and storage condition is 37 (n=37).

Were the data analyzed by normality tests? Insert this information in topic 2.5.

Review the results abbreviation. Some abbreviations are explained in the abstract only.

In the text, significant statistical differences are presented, however, these differences are not represented in tables or figures.

I do not agree with the table and figure representation of the same results (dimension accuracy and ultimate tensile strength). This information is duplicated, and no statistical representations are presented. Specifically, for table 2 and figures 5 and 6.

A poor discussion of ultimate tensile strength results was provided. Explain the mechanism of action of deterioration caused by thermal cycling. What is clinically observed?

References need to be standardized in accordance with Polymers guideline. Only 6 references of 24 are current. The literature needs to be updated. Studies with native language (not English) are inadvisable.

Author Response

Please see the details from the attachmen.

Round 2

Reviewer 2 Report

Dear authors, congratulations for your study. The reviews are adequate, so I'm considering this study for publication.

Author Response

Thank you for your invaluable input and the work done in order to increase our manuscripts' quality!

please see the revise version from the attachment
